# How far back do we need to look to capture diagnoses in electronic health records? A retrospective observational study of hospital electronic health record data

Jadene Lewis,[1,2] Felicity Evison ,[1,2] Rominique Doal,[1,2] Joanne Field,[3] Suzy Gallier,[1,2] Steve Harris,[4,5] Peta le Roux,[3] Mohammed Osman,[6,7] Chris Plummer,[3,7] Elizabeth Sapey ,[1,8] Mervyn Singer,[4,9] Avan A Sayer,[6,7] Miles D Witham ,[6,7] The ADMISSION Research Collaborative

For numbered affiliations see end of article.

**Correspondence to**
Professor Miles D Witham;
miles.witham@newcastle.ac.uk

## ABSTRACT

**Objectives** Analysis of routinely collected electronic health data is a key tool for long-term condition research and practice for hospitalised patients. This requires accurate and complete ascertainment of a broad range of diagnoses, something not always recorded on an admission document at a single point in time. This study aimed to ascertain how far back in time electronic hospital records need to be interrogated to capture long-term condition diagnoses.

**Design** Retrospective observational study of routinely collected hospital electronic health record data.

**Setting** Queen Elizabeth Hospital Birmingham (UK)-linked data held by the PIONEER acute care data hub.

**Participants** Patients whose first recorded admission for chronic obstructive pulmonary disease (COPD) exacerbation (n=560) or acute stroke (n=2142) was between January and December 2018 and who had a minimum of 10 years of data prior to the index date.

**Outcome measures** We identified the most common International Classification of Diseases version 10-coded diagnoses received by patients with COPD and acute stroke separately. For each diagnosis, we derived the number of patients with the diagnosis recorded at least once over the full 10-year lookback period, and then compared this with shorter lookback periods from 1 year to 9 years prior to the index admission.

**Results** Seven of the top 10 most common diagnoses in the COPD dataset reached >90% completeness by 6 years of lookback. Atrial fibrillation and diabetes were >90% coded with 2–3 years of lookback, but hypertension and asthma completeness continued to rise all the way out to 10 years of lookback. For stroke, 4 of the top 10 reached 90% completeness by 5 years of lookback; angina pectoris was >90% coded at 7 years and previous transient ischaemic attack completeness continued to rise out to 10 years of lookback.

**Conclusion** A 7-year lookback captures most, but not all, common diagnoses. Lookback duration should be tailored to the conditions being studied.

## STRENGTHS AND LIMITATIONS OF THIS STUDY

⇒ We analysed common conditions associated with two different index conditions underlying hospital admission.
⇒ Our mature, large hospital dataset enabled us to study a variety of conditions and analyse longitudinal trends over 10 years.
⇒ Less common associated conditions were not studied and may require different lookback periods for effective capture.
⇒ Our analysis was confined to a single electronic hospital record and the required lookback periods could differ between hospitals.

## BACKGROUND

Multiple long-term conditions (MLTCs) (multimorbidity) are an important but understudied challenge to healthcare systems.[1] MLTCs, defined as two or more long-term health conditions, are common with an estimated global prevalence of 37%.[2] The prevalence in old age is even higher, reaching 80% by the age of 80 years in the UK.[3] Much of the research on MLTCs to date has used large population-based datasets or has focused on primary care data; there has been comparatively little work on MLTCs within the hospital environment.[4] Understanding the associations, consequences and pathways of care for MLTCs within hospital is critical if we are to understand how best to prevent the adverse consequences of MLTCs and deliver better care for hospitalised patients.[5]

Analysis of routinely collected electronic healthcare records (EHRs) affords the opportunity to study MLTCs at scale, across a wide range of patient groups. Use of routine data

enables inclusion in analyses of those who would not be able to join traditional consented studies, such as those with dementia or with critical illness. The accurate identification of which long-term conditions affect hospitalised patients is central to the study of MLTCs in patients admitted to hospital. In the UK, diagnoses within EHRs are recorded at discharge, but can also be recorded during admissions. The quality and completeness of diagnostic coding are highly variable however,[6 7] and examining diagnoses coded during a single admission may not provide a complete picture of all diagnoses affecting an individual.

Aggregating diagnoses from a series of hospital admissions provides an opportunity to improve the completeness of recording, but it is unclear how far back through an EHR researchers should look. Different EHRs have been in existence for different amounts of time, and comparison between EHRs may introduce bias if different lookback times introduce differences in the completeness of diagnostic recording. Understanding the completeness of diagnostic recording for a given lookback period would help to avoid under-recording of diagnoses in both research analyses and clinical applications. This in turn would enable analysis teams to make informed judgements about what records and which conditions to study when analysing routinely collected electronic health data.

The aim of this analysis was therefore to investigate how far back we need to look in order to capture the majority of diagnoses documented from previous admissions, and whether there are differences between diagnoses in what the minimum lookback period should be.

## METHODS
### Hospital and data sources

The Queen Elizabeth Hospital Birmingham (QEHB) is a National Health Service (NHS) acute hospital providing secondary and tertiary care services to adults in a large urban and suburban catchment area in England, UK. The hospital has 1269 beds including 80 level 2/3 intensive care beds and an emergency department that assesses >300 patients per day. QEHB uses an EHR (PICS, Birmingham Systems) containing time-stamped, structured records including demographics, location, time of admission and discharge. It also contains details of all treatments and investigations, and physiological measurements such as pulse, blood pressure and respiratory rate derived from the National Early Warning Score version 2. Diagnoses are currently coded using the Systematised Nomenclature of Medicine–Clinical Terms (SNOMED) system, and historically used International Classification of Diseases version 10 (ICD-10). The EHR has been in place since 1999 and the Trust is a paperless environment for all care provision and planning.

For this analysis, we used two data extracts. We extracted data on all acute admissions between 1 January 2018 and 31 December 2018 with a discharge diagnosis of chronic obstructive pulmonary disease (COPD) exacerbation (as

a primary or secondary diagnosis). We also used SNOMED codes recorded in the EHR to resolve whether a diagnosis coded as 'COPD unspecified' was due to acute exacerbation. Using this time period enabled us to avoid any distorting effects from the COVID-19 pandemic. We took the date of the first hospital admission for COPD exacerbation in 2018 as the index date, and extracted data on age, sex and all hospital diagnoses coded by ICD-10 discharge codes within 10 years prior to the admission under study. For the second data extract, we repeated the procedure but extracted data on all acute admissions with a discharge diagnosis of stroke. We chose COPD exacerbation and stroke as exemplar conditions as they are common causes of admission to hospital and are associated with a substantial burden of comorbidity.

### Approval by PIONEER Data Trust Committee

Governance and linkage processes for PIONEER data have been published previously.[8]

### Patient and public involvement

The ADMISSION grant programme was co-designed with a panel of patients including patients with MLTCs. Two public co-applicants form part of the management team for the ADMISSION Programme, and studies within ADMISSION including this analysis are discussed with the ADMISSION Patient Advisory Group, who provide input on the design, interpretation and dissemination of results.

### Analyses

We derived diagnoses made during, or at discharge, from ICD-10 codes recorded in the EHR over the 10 years prior to the index admission date. Code lists used for each diagnosis are given in online supplemental material; the conditions selected were based on conditions previously used in UK Biobank,[9] modified to remove diagnoses less likely to be coded in hospital discharge data. Code lists were reviewed by clinicians in the research team prior to data extraction. We then examined the percentage of diagnoses that were captured by looking back from the date of index admission for increasing periods of time from 1 year to 10 years. We compared the proportion of patients with diagnoses captured for each lookback period with the number of diagnoses captured with 10 years of lookback. Only individuals with a full 10 years of EHR data were included in the analyses. We chose a threshold of 90% completeness against which to report results; this threshold was agreed by clinician authors as representing a balance between accuracy and achievability.

## RESULTS

We included 560 individuals in the COPD exacerbation analysis dataset (32 were excluded due to incomplete lookback data) and 2142 individuals in the acute stroke dataset (189 were excluded due to incomplete lookback data). A flow chart depicting selection of patient

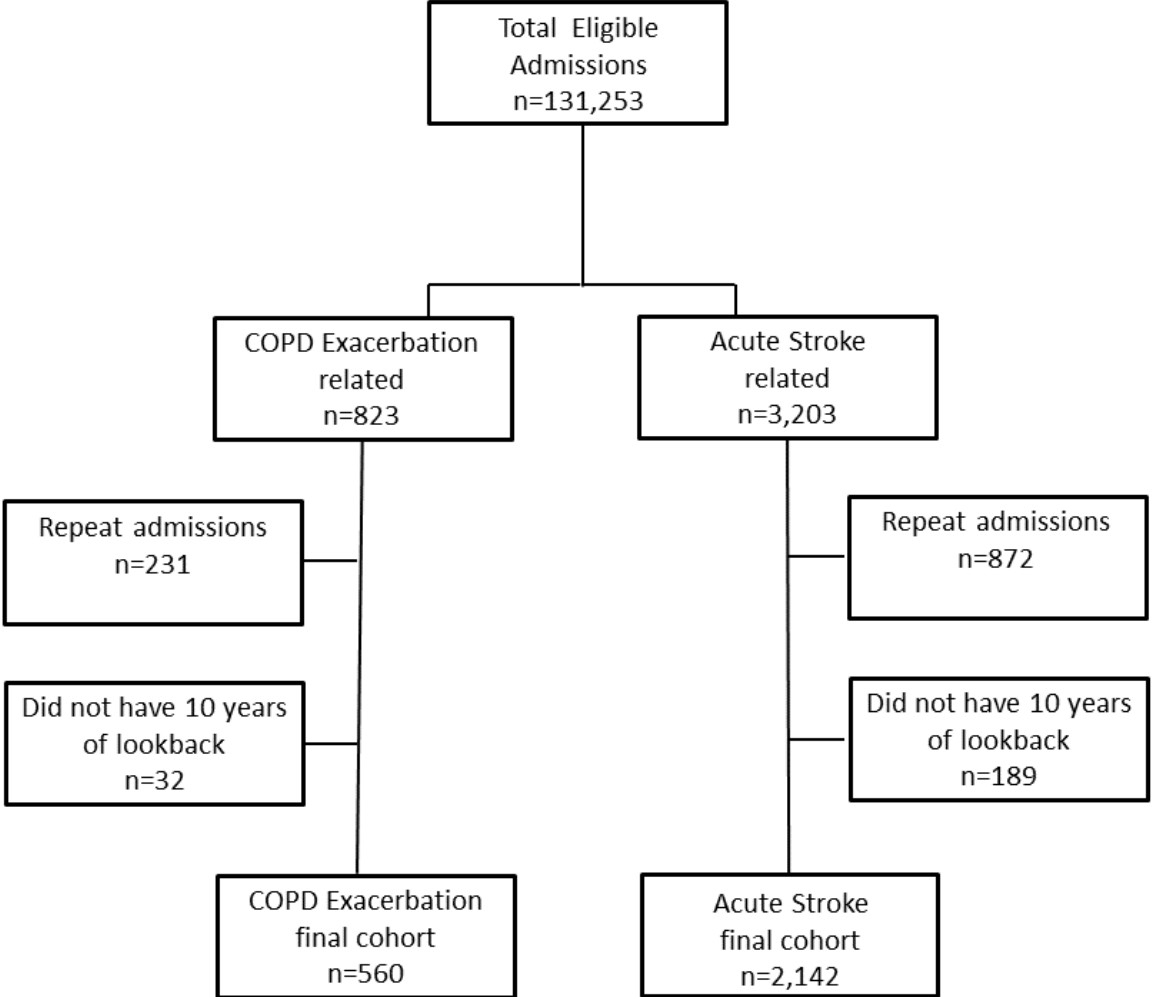

**Figure 1** Flow chart showing selection of patient episodes for inclusion in analysis. COPD, chronic obstructive pulmonary disease.

admissions is shown in figure 1. A total of 73 different diagnoses were recorded for those with COPD exacerbation (median three per individual) and a total of 80 different diagnoses were recorded for those with acute stroke (median six per individual). Table 1 shows details of individuals in each dataset, along with the top 10 most common diagnoses recorded in each dataset.

Figure 2 shows the percentage of each condition detected per year of lookback in the COPD dataset, using the number recorded over the full 10-year lookback period as reference. Seven of the top 10 reached 90% completeness by 6 years, but two conditions (asthma and hypertension) continued to show increases in coding completeness with increasing duration of lookback through to 10 years. Figure 3 shows similar data for the acute stroke dataset; there was considerable heterogeneity, from dementia (90% by 3 years) and myocardial infarction (90% by 4 years) through to diabetes mellitus (90% by 5 years) and angina pectoris (90% by 7 years). Previous transient ischaemic attack (TIA) coding completeness continued to increase through to 10 years with no flattening of the curve.

## DISCUSSION

Both cohorts in our study had a high prevalence of MLTCs, reinforcing the need for methodological studies in this area to ensure the robustness of analyses using routine data. Our results suggest that a 7-year lookback is sufficient to capture some but not all common diagnoses for patients admitted with COPD exacerbation or acute stroke. For most diagnoses investigated, the prevalence curves flatten out between 6 and 7 years, suggesting that additional years of lookback add few additional diagnoses. There were however notable exceptions, particularly hypertension and asthma in the COPD cohort, and previous TIA in the acute stroke cohort. For these diagnoses, the curves did not flatten at 10 years, suggesting that even longer periods of lookback may be required to gain complete diagnostic data. These data suggest that the duration of lookback needs to be tailored to the condition of interest.

Our results give a useful indication of what the minimum lookback time in hospital EHRs might need to be, but we are not able to ascertain why lookback time varies between diagnoses. Conditions which are the

**Table 1** Study population details

| | COPD dataset | | Stroke dataset | |
|---|---|---|---|---|
| N | 560 | | 2142 | |
| Mean age (years) (SD) | 71 (11) | | 73 (13) | |
| Female sex (%) | 296 (52) | | 1064 (50) | |
| Median number of comorbid conditions (IQR)* | 3 (1–5) | | 6 (4–8) | |
| Top 10 previously diagnosed conditions† | Name | Frequency (%) | Name | Frequency (%) |
| 1 | COPD | 560 (100) | Stroke | 1729 (81) |
| 2 | Heart failure | 227 (41) | Atrial fibrillation | 843 (39) |
| 3 | Asthma | 206 (37) | TIA | 832 (39) |
| 4 | Atrial fibrillation | 197 (35) | All diabetes | 804 (38) |
| 5 | Depression | 182 (33) | Heart failure | 631 (29) |
| 6 | Anxiety | 180 (32) | COPD | 525 (25) |
| 7 | All diabetes | 159 (28) | Myocardial infarction | 519 (24) |
| 8 | Hypertension | 151 (27) | Angina | 491 (23) |
| 9 | Myocardial infarction | 137 (24) | Dementia | 432 (20) |
| 10 | Angina | 126 (23) | Depression | 426 (20) |

*In addition to the index condition (COPD or stroke).
†Recorded at an admission prior to the index admission.
COPD, chronic obstructive pulmonary disease; TIA, transient ischaemic attack.

direct cause of hospital admission may be more likely to be recorded, and patients who have frequent hospital admissions may be more likely to have their associated diagnoses recorded. Conditions relevant to a particular specialty (for example, respiratory diagnoses when admitted under a respiratory team) may be more likely to

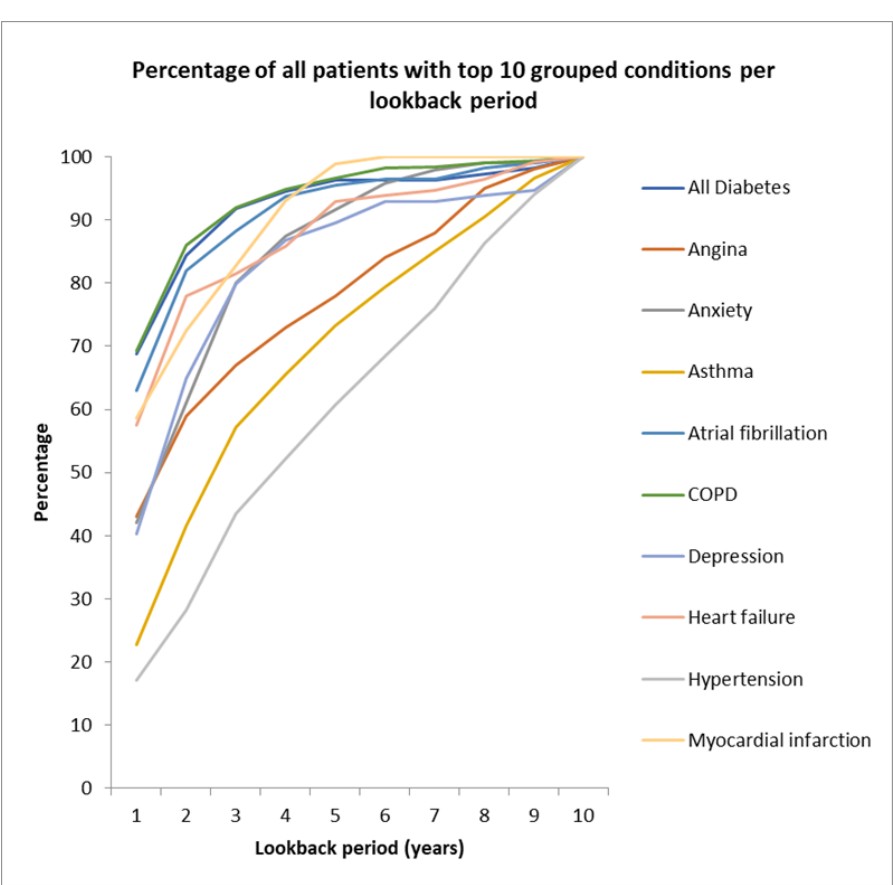

**Figure 2** COPD exacerbation top 10 previously diagnosed conditions. COPD, chronic obstructive pulmonary disease.

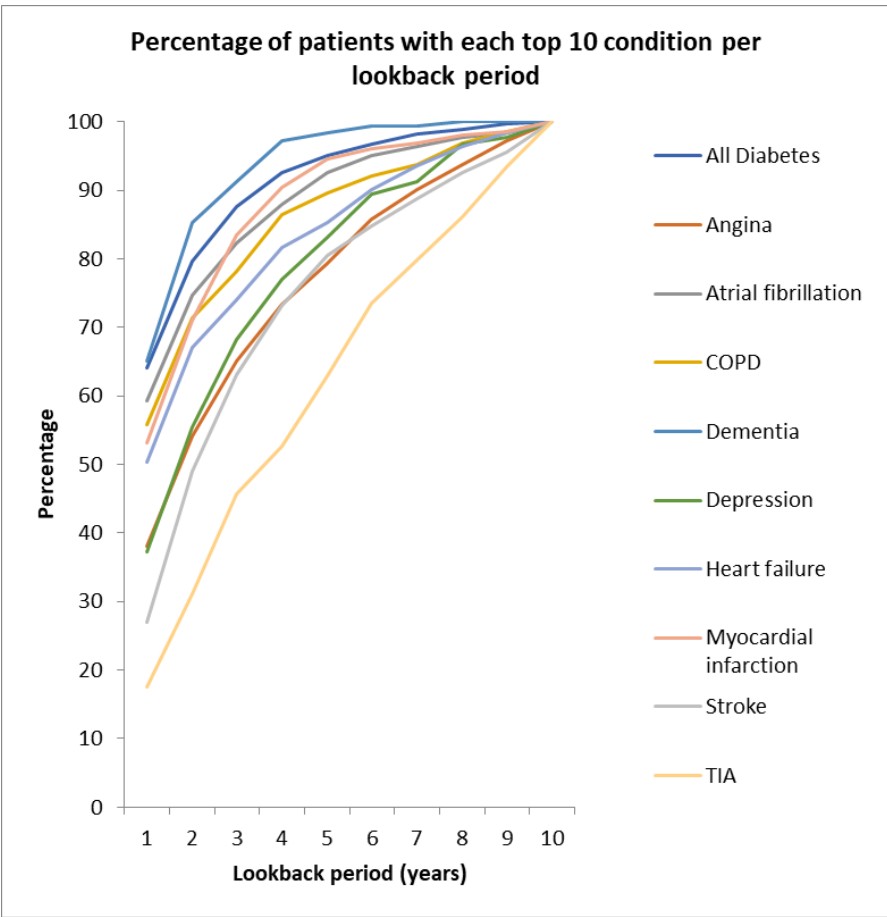

**Figure 3** Acute stroke top 10 previously diagnosed conditions. COPD, chronic obstructive pulmonary disease; TIA, transient ischaemic attack.

be recorded than those not associated with the admitting specialty. Further work is required to test these hypotheses. For some diagnoses, reliance on coded, structured diagnostic data is likely to remain inadequate as not all diagnoses are written down in hospital records. Even for those diagnoses that are written down, many are not recorded or coded in a structured manner.[10] Extracting these diagnoses requires additional work, for instance, by interrogating unstructured EHR data using techniques such as natural language processing,[11] by using additional data sources, for instance, prescribing records[12] or laboratory data,[13] and by linkage to alternative sources of diagnostic data (for instance, data from primary care).[14] Our results help to highlight diagnoses for which this work is particularly needed.

Strengths of our approach include the use of routinely collected electronic healthcare data and the interrogation of two common diagnostic reasons for hospital admission. Using routinely collected electronic health data for MLTC research enables inclusion of a wide range of patients. This includes groups excluded from traditional cohort studies because of inability to consent due to illness or cognitive impairment, or who belong to other groups underserved by research such as socioeconomically disadvantaged groups.[15]

Routine data also enable the use of very large sample sizes and can provide detailed longitudinal follow-up incorporating a wide range of conditions and outcomes. We were unable to look back further than 10 years without a reduction in the number of patients included in the analysis, and we have confined our analysis to the most common conditions associated with each index condition. Other less common conditions may require different lookback periods. Our choice of conditions was based on those previously analysed in UK Biobank, but UK Biobank does not use an exhaustive list of conditions and it is possible that other common and important conditions were not included in our analysis. Our choice of a 90% threshold for reporting analyses is to a degree arbitrary; different degrees of completeness may be appropriate for different analyses and only individual research or analysis teams can make a judgement as to what degree of completeness is appropriate for a particular analysis. Finally, our analysis is confined to a single hospital, and lookback times may vary across different healthcare organisations, which may vary in how well they code different diagnoses. Future research to replicate our work in different hospitals and in different index conditions would be useful to increase the generalisability of our findings.

In summary, while a 7-year lookback appears adequate to capture some common diagnoses associated with COPD exacerbation and acute stroke admissions, there is considerable variability between diagnoses. These findings will help both research teams and clinical teams decide what length of lookback they need to employ to ensure that capture of diagnoses is sufficiently complete for the intended purpose, whether this is for research, clinical audit or as part of routine healthcare delivery. For EHRs with only a short lookback window, this may mean that certain diagnoses are not amenable to accurate interrogation. The variability in lookback suggests that researchers need to evaluate diagnoses in the specific EHR system that they use to be sure that diagnosis capture is suitably complete for the conditions they wish to study.

**Author affiliations**
[1]PIONEER Hub, University of Birmingham, Birmingham, UK
[2]Health Informatics, University Hospitals Birmingham NHS Foundation Trust, Birmingham, UK
[3]Digital Services, Newcastle upon Tyne Hospitals NHS Foundation Trust, Newcastle upon Tyne, UK
[4]Critical Care Department, University College London Hospitals NHS Foundation Trust, London, UK
[5]Institute of Health Informatics, University College London, London, UK
[6]AGE Research Group, Translational and Clinical Research Institute, Faculty of Medical Sciences, Newcastle University, Newcastle upon Tyne, UK
[7]NIHR Newcastle Biomedical Research Centre, Newcastle upon Tyne Hospitals NHS Foundation Trust; Cumbria, Northumberland, Tyne and Wear NHS Foundation Trust and Newcastle University, Newcastle upon Tyne, UK
[8]Institute of Inflammation and Ageing, University of Birmingham, Birmingham, UK
[9]Bloomsbury Institute of Intensive Care Medicine, University College London, London, UK

**Acknowledgements** This work used data provided by patients and collected by the NHS as part of their care and support. MDW, CP and AAS acknowledge support from the National Institute for Health and Care Research (NIHR) Newcastle Biomedical Research Centre based at Newcastle upon Tyne Hospitals NHS Foundation Trust, Cumbria, Northumberland, Tyne and Wear NHS Foundation Trust and Newcastle University.

**Collaborators** ADMISSION Research Collaborative Consortium members: Avan Aihie Sayer (AGE Research Group, Translational and Clinical Research Institute, Faculty of Medical Sciences, Newcastle University, Newcastle upon Tyne, UK; NIHR Newcastle Biomedical Research Centre, Newcastle upon Tyne Hospitals NHS Foundation Trust, Cumbria, Northumberland, Tyne and Wear NHS Foundation Trust and Newcastle University, Newcastle upon Tyne, UK); Victoria Bartle (Public Co-Investigator, ADMISSION Research Collaborative, Newcastle upon Tyne, UK); Rachel Cooper (AGE Research Group, Translational and Clinical Research Institute, Faculty of Medical Sciences, Newcastle University, Newcastle upon Tyne, UK; NIHR Newcastle Biomedical Research Centre, Newcastle upon Tyne Hospitals NHS Foundation Trust, Cumbria, Northumberland, Tyne and Wear NHS Foundation Trust and Newcastle University, Newcastle upon Tyne, UK); Heather J Cordell (Population Health Sciences Institute, Faculty of Medical Sciences, Newcastle University, Newcastle upon Tyne, UK); Ray Holding (Public Co-Investigator, ADMISSION Research Collaborative, Newcastle upon Tyne, UK); Tom Marshall (Institute of Applied Health Research, University of Birmingham, Birmingham, UK); Fiona E Matthews (Population Health Sciences Institute, Faculty of Medical Sciences, Newcastle University, Newcastle upon Tyne, UK); Paolo Missier (School of Computing, Newcastle University, Newcastle upon Tyne, UK); Ewan Pearson (Division of Population Health and Genomics, Ninewells Hospital and School of Medicine, University of Dundee, Dundee, UK); Chris Plummer (Digital Services, Newcastle upon Tyne Hospitals NHS Foundation Trust, Newcastle upon Tyne, UK; NIHR Newcastle Biomedical Research Centre, Newcastle upon Tyne Hospitals NHS Foundation Trust, Cumbria, Northumberland, Tyne and Wear NHS Foundation Trust and Newcastle University, Newcastle upon Tyne, UK); Sian Robinson (AGE Research Group, Translational and Clinical Research Institute, Faculty of Medical Sciences, Newcastle University, Newcastle upon Tyne, UK; NIHR Newcastle Biomedical Research Centre, Newcastle upon Tyne Hospitals NHS Foundation Trust, Cumbria, Northumberland, Tyne and Wear NHS Foundation Trust and Newcastle University, Newcastle upon Tyne, UK); Elizabeth Sapey (PIONEER Hub, University of Birmingham, Birmingham, UK and Health Informatics, University Hospitals Birmingham NHS Foundation Trust, Birmingham, UK; Institute of Inflammation and Ageing, University of Birmingham, Birmingham, UK); Mervyn Singer (University College London Hospitals NHS Foundation Trust, London, UK and Bloomsbury Institute for Intensive Care Medicine, London, UK); Thomas Scharf (Population Health Sciences Institute, Faculty of Medical Sciences, Newcastle University, Newcastle upon Tyne, UK); James Wason (Biostatistics Research Group, Population Health Sciences Institute, Newcastle University, Newcastle upon Tyne, UK); Miles D Witham (AGE Research Group, Translational and Clinical Research Institute, Faculty of Medical Sciences, Newcastle University, Newcastle upon Tyne, UK; NIHR Newcastle Biomedical Research Centre, Newcastle upon Tyne Hospitals NHS Foundation Trust, Cumbria, Northumberland, Tyne and Wear NHS Foundation Trust and Newcastle University, Newcastle upon Tyne, UK).

**Contributors** Conception and design—JL, MDW, CP, ES, MS and AAS. Data curation and analysis—JL, FE, RD, SG, JF, SH, PIR and MO. Interpretation of results—MDW, AAS, FE and ES. Manuscript draft—JL and MDW. Critical revision of the manuscript—all authors. MDW is the guarantor for the work.

**Funding** This research was conducted as part of the ADMISSION Research Collaborative, funded by the Strategic Priority Fund 'Tackling multimorbidity at scale' Programme (grant number MR/V033654/1). This funding is delivered by the Medical Research Council and the National Institute for Health and Care Research in partnership with the Economic and Social Research Council and in collaboration with the Engineering and Physical Sciences Research Council.

**Disclaimer** The views expressed in this publication are those of the authors and not necessarily those of UK Research and Innovation, the National Institute for Health and Care Research, or the Department of Health and Social Care.

**Competing interests** ES is director of the PIONEER acute data hub, funded by Health Data Research-UK. The other authors have no conflicts of interest to declare.

**Patient and public involvement** Patients and/or the public were involved in the design, or conduct, or reporting, or dissemination plans of this research. Refer to the Methods section for further details.

**Patient consent for publication** Not required.

**Ethics approval** This study involves human participants and was approved under umbrella ethics and Confidentiality Advisory Group (CAG) approvals which underpin the work of PIONEER, the Health Data Research (HDR)-UK Hub in Acute Care (East Midlands–Derby Research Ethics Committee reference: 20/EM/0158 and CAG reference: 20/CAG/0084). The Section 251 support via the NHS Health Research Authority CAG gives permission for the use of routinely collected clinical data for research without individual participant consent. As part of this process, the data access request and specification were reviewed and approved by the PIONEER Data Trust Committee, comprising a group of patients and members of the public who provide oversight of all data access decisions within PIONEER.

**Provenance and peer review** Not commissioned; externally peer reviewed.

**Data availability statement** Data may be obtained from a third party and are not publicly available. The data that support the findings of this study are not openly available due to reasons of sensitivity. Data may be accessed on request to the HDR-UK PIONEER acute data hub on provision of permission from the PIONEER Data Trust Committee and provision of a data access agreement. Data are located in controlled access data storage at the PIONEER acute data hub.

## ORCID iDs

Felicity Evison http://orcid.org/0000-0002-9378-7548
Elizabeth Sapey http://orcid.org/0000-0003-3454-5482
Miles D Witham http://orcid.org/0000-0002-1967-0990

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
