## [Reviewer comments · BMJ Open]

ARTICLE DETAILS

TITLE (PROVISIONAL)	How far back do we need to look to capture diagnoses in electronic health records? An retrospective observational study of hospital electronic health record data
AUTHORS	Witham, Miles; Lewis, Jadene; Evison, Felicity; Doal, Rominique; Field, Joanne; Gallier, Suzy; Harris, Steve; le Roux, Peta; Osman, Mohammed; Plummer, Chris; Sapey, Elizabeth; Singer, Mervyn; Sayer, Avan; Research Collaborative, The ADMISSION

VERSION 1 – REVIEW

REVIEWER	de Lange, Melanie University of Bristol, MRC Integrative Epidemiology Unit
REVIEW RETURNED	24-Oct-2023

GENERAL COMMENTS	General comments: I think multimorbidity is a really important topic for researchers to be addressing. Our healthcare system is set up to deal with individual diseases when the reality is that most patients, particularly as they age, have multiple health conditions which all need to be considered together and as a whole. I also think that EHRs offer a huge, largely untapped resource for understanding and improving population health. On the surface this appears a relatively simple study but as someone who works with EHRs I appreciate the huge amount of data wrangling that goes on to make this look simple. Methodologically it is sound. I just have suggestions of how the write-up could be improved. See attached word doc for detailed comments.
--

REVIEWER	Gan, Ziming University of Chicago Division of the Physical Sciences, Statistics
REVIEW RETURNED	29-Oct-2023

GENERAL COMMENTS	Overall, I find your research question to be clearly defined and quite interesting: "How far back do we need to look to capture diagnoses." However, I believe there are several areas where the paper can be strengthened: 1. Statistical Definition of 'Sufficient Lookback': While your paper presents the observation that "90% of conditions were recorded by 6 years of lookback" and that "seven of the top 10 reached 90% completeness by 6 years," I would recommend that you provide a more formal statistical framework to determine what constitutes a 'sufficient lookback' period. This will not only enhance the generalizability of your findings but also make the methodology more transparent to your readers.2. Comparative Analysis of Lookback Periods: You currently focus on the minimum lookback period for COPD exacerbation and acute stroke admissions. To further enrich your paper, consider extending this analysis to include a wider range of diseases and diagnoses. By comparing lookback periods among
--

	different diseases, you could offer a more comprehensive perspective on the temporal patterns of diagnoses in healthcare records. 3. Background and Rationale: Your paper could benefit from a more detailed discussion in the background section to justify the significance of finding the minimum lookback period. Explaining why it doesn't work to examine more historical codes, such as those codified in the past twenty years, would provide a stronger foundation for your research. This context can help readers better understand the implications of your work.
--	---

REVIEWER	Dahia, Simranjeet Singh The University of Adelaide
REVIEW RETURNED	01-Nov-2023

GENERAL COMMENTS	The paper addresses an important research question related to the use of electronic health records in understanding long-term conditions. It provides valuable insights into the variable completeness of diagnostic data over different look back periods. However, there are some points to consider:  1. The rationale for choosing COPD exacerbation and acute stroke as the focus of this study is not clearly stated. Providing a succinct justification for this choice would enhance the clarity of the research objectives. 2. In the results section, it was mentioned that diagnoses are captured within 7 years for COPD and 5 years for acute stroke. It's unclear how this conclusion was reached, as it appears to contradict the earlier mentioned 6-year timeframe for COPD. Clarification on the methodology and reasoning behind this discrepancy is needed. 3. It's important to acknowledge that the selection of conditions was based on those previously used by the UK Biobank, which may result in the omission of important conditions not included in their dataset. This limitation should be discussed to highlight potential gaps in the analysis. 4. The study extracted the top 10 previously diagnosed conditions associated with each indexed condition, but the percentage of patients with these conditions along with COPD/stroke is not presented. Including this information would provide additional insights and enhance the value of the results. 5. The study was conducted in one hospital, and it's acknowledged that the number of look back years required may vary across hospitals and electronic health records. To strengthen the robustness of the findings, conducting a similar analysis at another hospital using the same EHR and comparing the results between hospitals is recommended. 6. The paper does not explore the reasons for the observed variability in completeness across diagnoses. Further discussion or hypotheses regarding this variability would have been valuable. 7. The language in the manuscript is occasionally informal, and there are instances where articles are missing before nouns. A thorough proofreading of the article is needed to ensure a consistent and formal tone throughout the manuscript. I would like to thank Dr. Laalithya Konduru, College of Medicine and Public Health, Flinders University, Adelaide, Australia for her valuable inputs in completing this Peer Review.
--

VERSION 1 – AUTHOR RESPONSE

Reviewer 1

General comments:

I think multimorbidity is a really important topic for researchers to be addressing. Our healthcare system is set up to deal with individual diseases when the reality is that most patients, particularly as they age, have multiple health conditions which all need to be considered together and as a whole.

I also think that EHRs offer a huge, largely untapped resource for understanding and improving population health.

On the surface this appears a relatively simple study but as someone who works with EHRs I appreciate the huge amount of data wrangling that goes on to make this look simple. Methodologically it is sound. I just have suggestions of how the write-up could be improved.

- *Thank you for your comment acknowledging the importance of the topic and the potential for EHR-based research to enable progress in this area*

Detailed feedback/potential areas to make it better

Title

Really like that you have a question as the title. I wonder if the subtitle could be a bit more informative? e.g. mention Queen Elizabeth Hospital Birmingham

- *Thankyou for this suggestion; we have amended the short title to include the hospital name (page 1)*

Abstract

I think your abstract needs to be more detailed as it didn't give me a great deal of insight into what you had done. I had to read the rest of the paper to confirm that what I guessed you had done was correct. I think it needs to be fleshed out with more detail.

Objectives: Can you put one sentence explaining *why* you want to ascertain how far back in time to go (you can summarise what you've put in the intro)?

Setting: Putting "The Queen Elizabeth Hospital Birmingham, UK" might give readers a better idea that you are using data from one UK hospital.

Participants: "Patients whose first recorded admission for COPD exacerbation (n=xxxx) or acute stroke (n=xxxx) was between January and December 2018 and who had a minimum of 10 years of data prior to the index date".

Outcome measures: "We identified the most common ICD-10 coded diagnoses received by COPD and acute stroke patients separately....."

- *We have made the suggested changes to the abstract (page 3-4)*

Strengths & weaknesses

I think bullet point 2 would be better as something like this: Our mature, large hospital dataset enabled us to study a variety of multimorbidities and analyse longitudinal trends over ten years.

- *We have amended bullet point 2 as suggested (page 4)*

Intro

Nice & succinct and gives a good rationale for why you need to look back in time. Are there any stats on the proportion of people of certain age groups with multimorbidity that you could add to your first paragraph to help give an idea of the scale of the problem you are ultimately trying to address?

A quick google search found this: [https://www.thelancet.com/journals/eclinm/article/PIIS2589-5370\(23\)00037-8/fulltext#:~:text=The%20subgroup%20study%20highlights%20that,CI%20%3D%2044.1%E2%80%9358.0%25\).](https://www.thelancet.com/journals/eclinm/article/PIIS2589-5370(23)00037-8/fulltext#:~:text=The%20subgroup%20study%20highlights%20that,CI%20%3D%2044.1%E2%80%9358.0%25).)

- *We have added additional content on the scale of the challenge posed by multimorbidity, along with the suggested reference (page 5 para 1)*

Method

You mention NEWS2 and SNOMED-CT – I think you need to explain what these are.

- *We have now added explanation of these (page 6 para 2)*

Analyses

I'm used to seeing a separate section purely dedicated to codelist creation as in EHR research the code lists you use are critical to the scope & results of the study. I appreciate that there are fewer codes in ICD10 than primary care so it is possible to create a comprehensive code list without having to do an electronic (coded) search so I think what you have done is fine. However, I would want those codelists to be checked by a clinician. I imagine this is the case but I'd like it to be stated explicitly in the paper.

- *These codelists were developed and checked by clinicians (who are coauthors on the paper); we have now added a sentence about this (page 7 para 4)*

Thank you for providing the complete code lists in the appendix! This is so important for reproducibility. Also, so much effort goes in to creating code lists and there is a lot of duplication of effort. It is really useful for people to share their codelists so they can be used (at least as a starting point) by other researchers.

- *Thank you for this comment. We agree that reproducibility is important, particularly for this type of study*

In your method you've said you have said the codelists comprise of ICD10 codes but the COPD codelist in the appendix has a couple of SNOMED codes listed. Please can you mention /explain that in your method.

- *Thank you for this comment, we have amended the text with the information about why how we used the additional information for SNOMED codes in COPD (page 6 para 2)*

Not sure whether this will be possible at this late stage in this project but going forward, for reproducibility purposes, it would be really useful to provide the code used to run your analyses. This can just be as simple as uploading your code files to a github site and providing the link in the paper.

- *Although we are not able to do this for this particular project, we acknowledge that this is an increasingly important issue and we are working with PIONEER team colleagues to find a generalizable way of doing this for future analyses using PIONEER data.*

Results

In the results section you've said that you excluded some participants due to incomplete lookback data. I am wondering whether it might be possible to have some kind of participant flow diagram (either in the methods or results section) outlining how

participants were included/excluded from each of your datasets. I don't know how your data was extracted but do you have the total number of admissions in 2018 to put at the top of a flow chart?

I like your table 1 and the figures do a nice job of demonstrating that the proportion of diagnoses captured increases the further back in time you go and differs between diagnoses.

- *We have now added a flowchart as suggested (Fig 1)*

Discussion

At the start of the discussion I think it is worth mentioning the fact that you did find high rates of multimorbidity in your datasets (median of 3 for COPD and median of 6 for stroke) as it links back to your intro where you are justifying the rationale for the study.

- *We have added a line on this as suggested (page 9 para 1)*

You've listed the use of routinely collected electronic healthcare data as a strength. Can you explain *why* this a strength? E.g. Sample size, longitudinal data, able to capture a large range of comorbidities?

- *We have expanded on our explanation of routine data as a strength as suggested (page 9 para 3)*

You've said 'our analysis is confined to a single electronic health record'. I think this sentence is a bit confusing. Instead I think you need to say your study is confined to one hospital.

- *We have amended this line as suggested (page 10 para 1)*

After paragraph 2 I would be really interested to see a bit more discussion of what you consider the research/practical implications of this study to be e.g. how does knowing they need to look back over a certain period of time benefit researchers of multimorbidity? How will this benefit patients? I imagine you have done this study because you wanted to know the answer to your question, but what are you planning to do next with this information?

- *We have now added some additional discussion on this point (page 11 para 1)*

Reviewer 2:

1. Statistical Definition of 'Sufficient Lookback': While your paper presents the observation that "90% of conditions were recorded by 6 years of lookback" and that "seven of the top 10 reached 90% completeness by 6 years," I would recommend that you provide a more formal statistical framework to determine what constitutes a 'sufficient lookback' period. This will not only enhance the generalizability of your findings but also make the methodology more transparent to your readers.

- *We have now clarified why we selected 90% completeness to report in this analysis (page 8 para 1). There is no formal framework for deriving this threshold; this is an arbitrary figure (as any selection would be) and we now comment further on this in the Discussion. We understand that different thresholds may be appropriate for different purposes however, and we have also highlighted this now in the Discussion (page 10 para 2)*

2. Comparative Analysis of Lookback Periods: You currently focus on the minimum lookback period for COPD exacerbation and acute stroke admissions. To further enrich your paper, consider extending this analysis to include a wider range of diseases and diagnoses. By comparing lookback periods among different diseases, you could offer a more comprehensive perspective on the temporal patterns of diagnoses in healthcare records.

- *Thank you for this comment. We agree that examining additional diseases would be of interest, but that work is outside the scope of the current analysis. We have added a comment in the discussion to highlight that this is an area that would benefit from further study (page 10 para 2)*

3. Background and Rationale: Your paper could benefit from a more detailed discussion in the background section to justify the significance of finding the minimum lookback period. Explaining why it doesn't work to examine more historical codes, such as those codified in the past twenty years, would provide a stronger foundation for your research. This context can help readers better understand the implications of your work.

- *We have now added some additional background on why understanding how far to look back is helpful to researchers in the background section (page 5 para 3) and have also revisited this in a conclusion section (page 11 para 1)*

Reviewer 3

1. The rationale for choosing COPD exacerbation and acute stroke as the focus of this study is not clearly stated. Providing a succinct justification for this choice would enhance the clarity of the research objectives.

- *We have added a line in the Methods section (page 7 para 1) to explain our choice of these two exemplar conditions*

2. In the results section, it was mentioned that diagnoses are captured within 7 years for COPD and 5 years for acute stroke. It's unclear how this conclusion was reached, as it appears to contradict the earlier mentioned 6-year timeframe for COPD. Clarification on the methodology and reasoning behind this discrepancy is needed.

- *We agree that this is slightly confusing. To clarify this and provide consistency with the abstract, we have amended the results paragraph in question (page 8 para 3) to omit mention of the lookback time to capture all diagnosis, choosing instead to focus on differences in lookback time for individual diagnoses in the COPD group and the stroke group.*

3. It's important to acknowledge that the selection of conditions was based on those previously used by the UK Biobank, which may result in the omission of important conditions not included in their dataset. This limitation should be discussed to highlight potential gaps in the analysis.

- *We agree that this is a potential limitation and we have now added a line about this in the Discussion (page 10 para 2)*

4. The study extracted the top 10 previously diagnosed conditions associated with each indexed condition, but the percentage of patients with these conditions along with COPD/stroke is not presented. Including this information would provide additional insights and enhance the value of the results.

- We have now added percentages to the frequencies of each condition shown in Table 1 as requested

5. The study was conducted in one hospital, and it's acknowledged that the number of look back years required may vary across hospitals and electronic health records. To strengthen the robustness of the findings, conducting a similar analysis at another hospital using the same EHR and comparing the results between hospitals is recommended.

- We agree that this would be desirable, but this work lies outside the scope of the current report. We have indicated that this could be a focus for future work in the Discussion (page 10 para 2)

6. The paper does not explore the reasons for the observed variability in completeness across diagnoses. Further discussion or hypotheses regarding this variability would have been valuable.

- We have added some additional discussion on this in the Discussion section (page 9 para 2)

7. The language in the manuscript is occasionally informal, and there are instances where articles are missing before nouns. A thorough proofreading of the article is needed to ensure a consistent and formal tone throughout the manuscript.

- We have gone through the manuscript making edits to grammar and style to address these points

VERSION 2 – REVIEW

REVIEWER	de Lange, Melanie University of Bristol, MRC Integrative Epidemiology Unit
REVIEW RETURNED	03-Jan-2024

GENERAL COMMENTS	You have completely addressed all of my comments. Your revised manuscript reads better and is much more comprehensive in terms of detailing the rationale, methods and implications of the study. Well done!
--

REVIEWER	Gan, Ziming University of Chicago Division of the Physical Sciences, Statistics
REVIEW RETURNED	17-Jan-2024

GENERAL COMMENTS	The investigation of this topic holds significant meaning. The inclusion of comprehensive code lists in the appendix ensures the reproducibility of the
---

	research. The authors have thoroughly examined the varying requirements of lookback duration for different purposes. Well-written!
REVIEWER	Dahia, Simranjeet Singh The University of Adelaide
REVIEW RETURNED	21-Jan-2024
GENERAL COMMENTS	The authors have effectively addressed the raised points, especially those from my previous review. In this round, my suggestions mainly pertain to refining the language for increased conciseness in the revised sections. For example, "Using routinely-collected electronic health data for MLTC research enables analyses to include a wide range of patients, many of whom are excluded from traditional cohort studies because of inability to consent due to illness or cognitive impairment, or who belong to other groups underserved by research such as socioeconomically disadvantaged groups." can be rewritten as: "Leveraging electronic health data for MLTC research broadens patient inclusion, overcoming barriers seen in traditional cohort studies due to consent challenges arising out of illness and cognitive impairment, and underrepresentation of socioeconomically disadvantaged groups." Please note that the above is only as an example; the entire paper needs to be revised for readability and conciseness.

VERSION 2 – AUTHOR RESPONSE

Thankyou for passing on the reviewer comments on our revised manuscript. We are pleased that reviewers 1 and 2 are happy with the changes we have made.

We note reviewer 3's request to simplify the structure of our writing in places; we have gone through the manuscript and amended our choice of wording for improved readability, and we have shortened or broken up a number of longer sentences. The specific example given by reviewer 3 is an example of this – we have amended our text but elected to keep some of our original wording as it better reflects our meaning than the suggested edit (for instance, underserved is preferred to underrepresented when discussing inclusion in research as per the UK NIHR INCLUDE guidance)

We hope that these changes address the point raised about readability and we look forward to your decision on our revised manuscript